# Aging of Bone Marrow Mesenchymal Stromal Cells: Hematopoiesis Disturbances and Potential Role in the Development of Hematologic Cancers

**DOI:** 10.3390/cancers13010068

**Published:** 2020-12-29

**Authors:** Fulvio Massaro, Florent Corrillon, Basile Stamatopoulos, Nathalie Meuleman, Laurence Lagneaux, Dominique Bron

**Affiliations:** 1PhD Program in Clinical and Experimental Medicine, University of Modena and Reggio Emilia, 41121 Modena, Italy; fulvio.massaro@bordet.be; 2Department of Hematology, Jules Bordet Institute (ULB), 1000 Brussels, Belgium; nathalie.meuleman@bordet.be (N.M.); dominique.bron@bordet.be (D.B.); 3Laboratory of Clinical Cell Therapy, ULB-Research Cancer Center (U-CRC), Jules Bordet Institute, Université Libre de Bruxelles (ULB), 1000 Brussels, Belgium; basile.stamatopoulos@ulb.ac.be (B.S.); laurence.lagneaux@bordet.be (L.L.)

**Keywords:** mesenchymal stromal cells, aging, bone marrow niche, hematopoiesis, hematologic malignancies, inflammation, inflammaging

## Abstract

**Simple Summary:**

As for many other cancers, the risk of developing hematologic malignancies increases considerably as people age. In recent years, a growing number of studies have highlighted the influence of the aging microenvironment on hematopoiesis and tumor progression. Mesenchymal stromal cells are a major player in intercellular communication inside the bone marrow microenvironment involved in hematopoiesis support. With aging, their functions may be altered, leading to hematopoiesis disturbances which can lead to hematologic cancers. A good understanding of the mechanisms involved in mesenchymal stem cell aging and the consequences on hematopoiesis and tumor progression is therefore necessary for a better comprehension of hematologic malignancies and for the development of therapeutic approaches.

**Abstract:**

Aging of bone marrow is a complex process that is involved in the development of many diseases, including hematologic cancers. The results obtained in this field of research, year after year, underline the important role of cross-talk between hematopoietic stem cells and their close environment. In bone marrow, mesenchymal stromal cells (MSCs) are a major player in cell-to-cell communication, presenting a wide range of functionalities, sometimes opposite, depending on the environmental conditions. Although these cells are actively studied for their therapeutic properties, their role in tumor progression remains unclear. One of the reasons for this is that the aging of MSCs has a direct impact on their behavior and on hematopoiesis. In addition, tumor progression is accompanied by dynamic remodeling of the bone marrow niche that may interfere with MSC functions. The present review presents the main features of MSC senescence in bone marrow and their implications in hematologic cancer progression.

## 1. Introduction

Cancers and aging are closely linked [1]. Indeed, in most organisms, aging is accompanied by multiple alterations at the cellular, tissue and systemic levels. All of these alterations provide fertile ground for the development and progression of tumors, as evidenced by the shared hallmarks of aging and cancers [2,3]. Although the intrinsic processes leading to cell transformation from a normal cell into a tumor cell are now well known, it is also commonly accepted that the microenvironment surrounding cells and the interactions between malignant cells and this microenvironment play crucial roles in tumor development and growth. Malignant hemopathies represent no exception: they comprise a wide collection of disorders, all originating from cells of the bone marrow (BM) and the lymphatic system and accounting for almost 230,000 new cases every year in Europe [4].

The homeostasis and maintenance of BM cells and the immune system require continuous renewal of all types of blood cells. This function is ensured in the BM by hematopoietic stem cells (HSCs) that can differentiate into myeloid progenitors, giving rise to erythrocytes, platelets, granulocytes, and monocytes, or into lymphoid progenitors, giving rise to B lymphocytes, T lymphocytes and NK cells. The function and regulation of HSCs are supported by their close environment, the BM niche [5,6,7]. A growing number of studies point to a clear link between aging, remodeling of the BM microenvironment and impairment of hematopoiesis, leading, among other things, to hematologic cancers [8,9]. The BM niche is a complex and dynamic network that is not yet fully understood and is regulated by a wide number of cell types: endothelial cells, mesenchymal stromal cells (MSCs), perivascular stromal cells, osteoblasts, sympathetic neurons, nonmyelinating Schwann cells, adipocytes and regulatory T cells.

MSCs are multipotent nonhematopoietic cells able to differentiate into osteoblasts, chondrocytes, adipocytes and fibroblasts [10,11,12]. They also secrete a wide variety of compounds, such as growth factors, antiapoptotic factors, angiogenic factors and several cytokines, and thus contribute to the regenerative process, wound healing, hematopoietic support and regulation of the immune response [13,14]. MSCs also produce a large amount of extracellular vesicles (EVs), small vesicles playing a major role in cell-to-cell communication. EVs transport different elements, such as proteins, lipids and microRNAs (miRNAs), to target cells and are involved in many biological functions of MSCs [15,16,17]. It is now known that the aging of MSCs alters their EV production and has a direct impact on their functions and differentiation capacities [18,19]. Aging is also associated with an increased incidence of hematologic malignancies such as chronic and acute leukemias, non-Hodgkin lymphomas and plasma cell disorders: the mean age at diagnosis is 65–70 years, and the incidence typically increases in groups of older subjects [4,20]. Due to their functions, MSCs are important actors in the tumoral microenvironment, but their exact role remains ambiguous. Indeed, different studies carried out to date show both a protumoral and an antitumoral function of MSCs, as reviewed by Galland and Stamenkovic in *The Journal of Pathology* [21]. Although the prominent role of MSCs in vivo seems to be participation in tumor progression, further studies will be necessary to obtain a deep understanding of their exact role inside the tumoral microenvironment.

In this review, we will start by highlighting the effects of aging on the functions of bone marrow mesenchymal stromal cells (BM-MSCs) inside the BM niche and their effects on hematopoiesis. Then, we will discuss the tumorigenic potential of BM-MSCs in the case of hematologic cancers.

## 2. The Role of BM-MSCs in BM and Hematopoiesis Alterations during Aging

As mentioned above, during the aging process, changes in HSCs and hematopoiesis disturbances occur. More precisely, the number of lymphoid progenitors decreases to the benefit of myeloid progenitors that increase but lose some of their functions [22,23]. The result is an alteration of the immune system, leading to an increased susceptibility to infections and to the development of autoimmune diseases and cancers. The aging of HSCs is due in part to cell-intrinsic factors, as reviewed by Mejia-Ramirez and Florian in *Haematologica* [24], but also by external signals from the aging microenvironment of which BM-MSCs are a part. In this section, we will discuss the main features of BM-MSC aging and its consequences on hematopoiesis and the inflammatory state of BM through the modifications of BM-MSC secretome, the imbalance of their immunomodulation properties and the imbalance between osteogenesis and adipogenesis leading to progressive replacement of bone by fat (Figure 1). We will also briefly examine the spatial and functional heterogeneity of BM-MSCs inside the BM niche and its change during aging.

### 2.1. Epigenetic and Secretome Modifications Associated with BM-MSC Aging

MSCs are multipotent cells with proliferative properties. However, similar to any normal cell, they can only undergo a limited number of cell divisions before entering a senescent state. Cellular senescence and its related cell cycle arrest were observed for the first time by Hayflick in long-term in vitro culture of human fibroblasts [25]. Since then, a wide variety of factors causing MSC senescence have also been described, such as oxidative stress [26], telomere attrition occurring during in vitro expansion [27] or unrepaired DNA damages [28]. Accumulation of senescent cells was also observed in several aged tissues, as it was well illustrated in a recent study evaluating the expression of p16 and p21, two markers of senescence, in organs from young or old donors [29]. An increased level of p21 was also observed in BM-MSCs from elderly people, suggesting that senescent BM-MSCs accumulate with physiological aging [30]. Nevertheless, some experiments studying MSC senescence do not use cells form elderly donors but rather in vitro stress-induced-senescence conditions such as long-term culture expansion or senescence induced by gamma irradiation. It is therefore necessary to remain cautious when comparing data concerning in vitro senescence with physiological aging.

Several pathways and actors implicated in MSC senescent cell cycle arrest have been identified: the well-established p53/p21 and p16/pRB pathways, as well as the AKT/mTOR pathway [31], JAK/STAT pathway [32], mitogen activated protein kinase p38MAP [33] and fibroblast growth factor FGF21 [34].

Senescence is accompanied by many cellular modifications at morphological and functional levels: the cells become larger and resistant to apoptosis, while the autophagy process decreases, and they are also subject to genetic and epigenetic modifications [35,36]. Epigenetic modifications are key components of the BM niche homeostasis and can contribute to age- and disease-associated MSC alterations. Modifications of MSC DNA methylation patterns and hypermethylated and hypomethylated CpG sites in several genomic loci have been observed upon aging and replicative senescence [37,38]. Some epigenetic regulators have been identified to participate in MSCs aging. The expression and activation of Sirt1, a NAD-dependent histone deacetylase, decrease with age and modify MSC proliferation and differentiation [39,40]. Interestingly, miR-199b-5p, which is predicted to target Sirt1, is deregulated in old BM-MSCs [41]. MSCs deficient in Sirt6, another histone deacetylase, displayed accelerated cellular senescence, dysregulated redox metabolism and increased sensitivity to oxidative stress [42]. In HSCs, identification of somatic mutations in the epigenetic regulators DNMT3, TET2 and ASXL1 is associated with an increased risk of developing hematologic cancers [43]. These mutations can occur as people age and their identification in healthy people is known as clonal hematopoiesis of indeterminate potential (CHIP). These three epigenetic regulators seem to be involved in MSC aging. In human umbilical cord blood-derived MSCs (UC-MSCs), the inhibition of DNMT1 and DNMT3b induces cellular senescence [44]. In a mouse model used to study age-related skeletal diseases, the expression of TET2 resulted decreased [45]. In addition, it has been shown in mice that a loss of ASXL1 or TET2 impairs BM-MSCs fate and their ability to support hematopoiesis [46,47]. Taken together, these two observations suggest that epigenetic modifications of BM-MSCs occurring during aging can contribute to hematopoiesis disturbances.

The alterations associated with BM-MSC senescence also lead to a deep modification of their secretome, making them adopt a new phenotype called the senescent-associated secretory phenotype (SASP) [35] (Figure 1A). This SASP is characterized by increased secretion of growth factors, proangiogenic factors, extracellular matrix remodeling factors and especially proinflammatory cytokines such as IL-1β, IL-6 and IL-8 [48,49,50,51,52]. It is now well known that EVs contribute greatly to the SASP and that senescence of MSCs has a strong impact on them: their secretion is increased while their size is reduced and their content is modified, especially in terms of miRNA [18,53]. For example, the activation of AKT in aged BM-MSCs leads to increased partitioning of miR-17 and miR-34a in EVs, which, upon transfer to HSCs, cause functional impairment via downregulation of autophagy-related genes [54]. Terlecki-Zaniewicz and colleagues suggested that EVs of dermal fibroblasts and their miRNAs act as cargo for novel members of the SASP that are selectively secreted or retained in cellular senescence [55]. Although there are no similar experimental data on MSCs, it is reasonable to assume this may also be applicable to them. Robbins suggested that senescent cell-derived EVs could function as pro-geronic factors [56]. The SASP participates in the establishment of the low-grade and chronic inflammation state observed during aging, called inflammaging [57,58].

In addition, proinflammatory cytokines induce the expression of other cytokines by BM-MSCs [59]. It has been reported that BM-MSCs secrete a significant amount of IL-6 in response to TNF-α and IFN-γ [60] and that this cytokine is secreted at higher levels by aged BM-MSCs [51]. IL-6 is a proinflammatory cytokine implicated in inflammaging or promoting tumor growth and metastasis formation. The serum of elderly patients with chronic disease or cancer usually contains more IL-6 than the serum of young healthy people [61]. The MSCs contribute thus, by this cytokine release, to the “inflammaging” process well described by Fulop and others in age-related diseases [62]. With TGF-β, TNF-α and GM-CSF, IL-6 also promotes the differentiation of HSCs towards the myeloid lineage [51,63]. Interestingly, a recent analysis of the secretome of adipose tissue-derived MSCs (AT-MSCs) showed that GM-CSF is more highly secreted by senescent MSCs obtained by successive passaging [64]. Other proinflammatory cytokines secreted by BM-MSCs could impair hematopoiesis. Indeed, although IFN-γ promotes stem cell factor (SCF) expression in mouse BM-MSCs, an important factor for the support of hematopoiesis, chronic exposure of BM-MSCs to this cytokine leads to a decrease in the total number of BM-MSCs, diminishing their hematopoietic support [65]. In a recent paper, Gnani and colleagues showed a clonogenic impairment of young hematopoietic stem and progenitor cells (HSPCs) due to the activation of a proinflammatory program when they are exposed to compounds secreted by aged BM-MSCs [52].

### 2.2. Imbalance between Pro- and Anti-Inflammatory Functions

In addition to their role as multipotent progenitor cells, MSCs are also endogenous regulators of inflammation capable of immunosuppressive or proinflammatory functions depending on environmental conditions [66]. Thus, they express several Toll-like receptors (TLRs) whose activation influences their immunologic properties. In normal nonsenescent MSCs, the activation of TLR3 or the presence of IFN-γ and TNF-α polarizes the MSCs towards an anti-inflammatory state that is able to negatively regulate the proliferation of T lymphocytes and NK cells through the secretion of nitric oxide synthase (NOS) and prostaglandin E2 (PGE2), respectively [67]. MSCs also promote the polarization of macrophages into the alternatively activated anti-inflammatory type 2 state (M2) at the expense of the classical proinflammatory type 1 state (M1) [68].

These observations suggest that within the aging and inflamed microenvironment of the BM, BM-MSCs may influence other immune actors to counteract inflammation. However, the studies described in literature to date do not point in this direction. For example, it has been shown that gamma-irradiated senescent BM-MSCs showed a lower capacity to migrate in response to proinflammatory signals and, at least in part, a lower inhibitory capacity towards T lymphocytes [69]. In addition, the priming of UC-MSCs with IFN-γ and TNF-α induces the phosphorylation of p38MAP kinase only in aged MSCs, which could in turn negatively regulate the COX2/PGE2 pathway and explain at least partially the reduction in the immunomodulatory capacity of aged MSCs [70]. Other studies have highlighted the impact of MSC aging on macrophage polarization. In mice, indirect coculture of macrophages with BM-MSCs using transwells induces differential gene expression in macrophages depending on the aging state of BM-MSCs. Thus, TNF-α and iNOS, two markers of M1 proinflammatory macrophages, are upregulated after coculture with aged BM-MSCs compared to coculture with young BM-MSCs, which instead induce the expression of IL-10 and ARG1, two markers of the M2 state [71]. More recently, in a lung injury murine model, Huang and colleagues showed that injections of EVs secreted by young human BM-MSCs but not those secreted by the oldest BM-MSCs are able to polarize macrophages towards the M2 state, reducing the severity of lung injury [72]. These differences between old and young EVs might be due to their miRNA content.

Taken together, these results suggest that the aging of MSCs impairs their ability to adopt an immunosuppressive phenotype in response to environmental stimulation (Figure 1B). Interestingly, TLR signaling involved in MSC immune polarization is also implicated in the proliferation of HSCs towards the myeloid lineage and in the migration of monocytes [73,74].

### 2.3. Imbalance between Osteogenesis and Adipogenesis

Bone tissue is a dynamic tissue undergoing constant remodeling throughout its lifetime. Its homeostasis is maintained by two complementary processes: the formation of new bone by osteoblasts and the resorption of old and damaged tissues by osteoclasts. BM-MSCs play an important role in this balance by being recruited at the bone-resorptive site through TGF-β1 signaling and by differentiating into osteoblasts [75]. However, during aging, bone resorption increases, and the bone density of the organism progressively decreases, leading to osteoporosis and increasing the risk of fractures [76]. The age-related changes in MSC differentiation potential have been studied by several groups in mice and humans. Although conflicting results have been reported, one cause for the imbalance between bone and adipose tissue occurring with aging could be due to a gradual loss of the ability of BM-MSCs to differentiate into osteoblasts, favoring differentiation into adipocytes (Figure 1C). A study using senescent BM-MSCs obtained after long-term culture showed an increased osteogenic differentiation potential after several passages [77]. However, other studies comparing BM-MSCs harvested from young and old donors have shown both a maintenance [78,79] or a decrease in the osteogenic differentiation of oldest BM-MSCs [30,80,81,82]. In a recent study, authors analyzed the transcriptional profile of freshly isolated BM-MSCs from young and old donors and showed the upregulation of genes implicated in the peroxisome proliferator-activated receptor (PPAR) signaling in the oldest group, suggesting a reinforcement of pro-adipogenic microenvironment with aging [83].

Several factors are implicated in the control of BM-MSC differentiation: RUNX2 and SP7 promote osteogenesis [84,85], while CEBPα, CEBPβ, CEBPγ and PPARγ promote adipogenesis [86]. There is a growing body of data highlighting the age-dependent control of these factors. In mice, it has been shown that FOXP1, a transcription factor interacting with CEBPβ, is downregulated during aging [87]. Similarly, CBFβ and MAF, two cofactors of RUNX2, are also downregulated with increasing age [88,89], while PPARγ is upregulated [90]. All of these signaling pathway modifications promote adipogenesis. The miRNA content of BM-MSCs and their EVs also seem to be involved in the imbalance between osteogenesis and adipogenesis. Indeed, it has been shown that aging and oxidative stress can alter the miRNA cargo of EVs, which in turn causes the suppression of cellular proliferation and osteogenic differentiation of BM stromal cells [91]. It has also been reported that miR-31a-5p level rises in aged BM-MSCs and appears to be involved in increasing adipogenesis and decreasing osteogenesis [92]. The decrease in osteogenic differentiation by BM-MSCs is accompanied by a reduced level of osteopontin secretion, which is known to negatively regulate the self-renewal of HSCs [93,94].

Adipocytes in BM impair hematopoiesis by diminishing the differentiation of hematopoietic progenitors towards the B lymphocyte lineage [95]. In a recent paper, Aguilar-Navarro et al. observed an increase of adipocytes in BM of elderly people associated with an increase of maturing myeloid cells and they proposed a contributive role for adipocytes in myeloid skewing [96]. Another study conducted on mice has been shown that aging is associated with the expansion of adipogenic potential of a stem cell-like subpopulation in the BM which, in turn, altered hematopoiesis through an excessive production of Dipeptidyl peptidase-4 [97].

By increasing the number of adipocytes inside the BM, the aging of BM-MSCs could also indirectly impact the inflammatory state of the BM niche. It is indeed well known today that adipose tissue participates in the production of a large amount of soluble factors and cytokines and that aging and metabolic diseases, like obesity, are correlated with an increase of its proinflammatory cytokine secretion [98].

### 2.4. Functional and Spatial Heterogeneity of BM-MSCs

MSCs represent a complex cell population characterised by specific localisation and functional heterogeneity that may be essential to their biological role. Several surface markers can be used to identify the different subpopulations of MSCs [99]. In BM, CD271 antigen can be used to identify a subset of BM-MSCs able to inhibit the proliferation of allogenic T-lymphocytes and presenting lympho-hematopoietic engraftment-promoting properties [100]. Most of HSCs are located in intimate cell-cell contact with these CD271+ MSCs [101]. A low or negative expression of platelet derived growth factor receptor alpha (PDGRF-α) by CD271+ MSCs is correlated with expression of key-genes for HSC supportive function and this expression is modulated according to the different phases of development of the organism [102]. CD271+ cells can be further divided in two cell subgroups with different localisation depending on the expression of CD146. CD146+ status defines MSC population located in the perivascular spaces while CD146− cells are found in the endosteal region [103]. These populations have different degree of maturity: CD146− MSCs are more mature whereas CD146+ cells retain plasticity. Their distribution varies with age: Maijenburg et al. showed a predominance of CD146+ subset in pediatric and fetal BM and suggested that variation in MSC subpopulations is a dynamic process that can change MSC functions during aging of the BM [104].

Other studies using a new method of single cell transcriptional analysis showed age-related changes in BM-MSCs composition. Duscher et al. identified an age-related depletion of a subpopulation characterized by a pro-vascular transcriptional profile [105]. More recently, Khong and colleagues identified a unique quiescent subpopulation exclusively present in MSCs from young donors and showed that this subpopulation was characterized by a higher expression of genes involved in tissue regeneration [106].

It has also been described the existence of two populations of MSCs with neural crest or mesoderm embryonic origins and particularly the neural crest has been proposed as a source of MSCs with specialized hematopoietic stem cell niche function [107]. Embryonic origin has also been shown to play an essential role in the age-related decrease in the functional capacities of BM-MSCs [108].

## 3. BM-MSCs and Hematologic Malignancies

BM-MSCs play a dual role in tumor cell growth in vitro and in vivo: they suppress tumor cell proliferation and inhibit tumor growth, but they also suppress tumor cell apoptosis and promote tumor growth [21,109]. We will examine these different mechanisms in the context of hematologic malignancies below, but it is also important to note that several studies have also highlighted the link between MSCs and metastatic process in solid tumors. In breast cancer, MSC activity through CCL5 release and Tac1 upregulation markedly increased tumoral metastatic capacity [110,111]. In neuroblastoma, differences in both qualitative and quantitative features of MSCs affect tumoral progression in BM [112]. A MSC subpopulation expressing stemness, endothelial and pericytic cell markers seems to impair neoplastic cells homing to BM in breast and prostate cancer models [113]. These findings, even if impossible to apply to hematologic malignancies, demonstrate that MSCs are implicated in the regulation of the interactions between neoplastic cells and BM niche.

### 3.1. Antitumoral Role of BM-MSCs in Hematologic Cancers

The role of the BM microenvironment in the pathogenic process of several hematologic malignancies has been demonstrated in several settings [114]. Concerning the specific role of MSCs, their action towards neoplastic clones is still unclear. Several studies have underlined the antiproliferative action of MSCs, almost all in in vitro experiments with oncohematologic cell lines, not primary patient cells. MSCs from BM, adipose tissue and umbilical cord were used, showing the same activity independent of tissue origin.

Cell cycle arrest is one of the most accepted mechanisms explaining MSC antineoplastic activity: an experiment conducted using murine B-cell lymphoma, acute lymphoblastic leukemia (ALL) and erythroleukemia cells reported reduced cell cycle proliferation and IL-10 levels in the presence of murine BM-MSCs [115]. Similar results were obtained using human BM-MSCs in coculture with chronic myeloid leukemia (CML), acute myeloid leukemia (AML) and T-ALL cell lines, showing significantly reduced cyclin D2 activity and subsequent G1 phase blockade [116]. The use of AT-MSCs was associated with the same results [117]. Both cell-to-cell contact and paracrine signals were used to explain the antiproliferative effect [118,119,120].

Some of these findings were confirmed in murine models, in the majority of cases utilizing human MSCs. An in vivo demonstration of the antitumoral effect derived from coculture of human MSCs with hematologic malignancy cell lines was reported in CML: AT-MSCs inhibited CML cell line proliferation by interfering with the Wnt pathway and β-catenin production, blocking cell cycle progression to the G1/S checkpoint through the production of Dickkopf-related protein (Dkk-1) [121]. Similar antineoplastic activity was also reported towards non-Hodgkin lymphoma cell lines in a mouse xenograft model: BM-MSCs were capable of slowing tumor size increases and determining extensive intralesional necrosis, probably in relation to reduced angiogenesis [122].

### 3.2. Protumoral Role of BM-MSCs in Hematologic Cancers

On the other hand, the antiapoptotic activity driven by MSCs towards neoplastic cells has been widely documented in in vitro oncohematologic models. Among the most important described mechanisms, direct contact between MSCs and neoplastic cells seems to play a crucial role in the regulation of specific signaling pathways, mostly conferring increased survival capacity. This evidence was reported in two studies using primary B-ALL and chronic lymphocytic leukemia (CLL) cells, in which direct contact with BM-MSCs resulted in reduced apoptosis of leukemic cells, with a particular advantage for CLL cells adhering to the MSC layer [123,124]. Our group described increased B cell lymphoma-2 (BCL-2) expression in CLL patient cells cocultured with BM-MSCs [125]. In B-ALL, similar antiapoptotic activity was related to Notch-3 and Notch-4 pathway activation and increased PGE2 secretion [126,127]. Another significant pathway seems to be related to phosphatidylinositol-3-kinase (PI3K)/protein kinase B (AKT)-BAD hyperexpression [119].

Increased tumor cell stemness could play a key role in the survival promotion of hematologic malignancies: a model of multiple myeloma (MM) cell lines cocultured with BM-MSCs was associated with the overexpression of Bruton tyrosine kinase (BTK) and increased mRNA and protein expression of NANOG, OCT4 and SOX2, key genes for stem cell self-renewal [128].

BM-MSCs could induce drug resistance, as reported in several studies performed in AML, ALL and CML settings. Concerning AML, the possibility of conferring resistance towards multiple key drugs, such as cytarabine and the majority of anthracyclines (mitoxantrone, idarubicin and doxorubicin), acting on several signaling pathways, such as nuclear factor (NF)-кB, c-Myc, and Notch, has been demonstrated [129,130,131]. Similarly, the resistance to cytarabine, methotrexate, idarubicin and dexamethasone in ALL seems to involve MSC activity on extracellular signal-regulated kinase (ERK), p21 and Jagged1/Notch1 pathways. TGF-β1 and CXCL12 were reported to be two major regulators implicated in AML proliferation and chemoresistance [132]. Interestingly, BM-MSCs are also related to resistance to biological drugs, such as the BCR/ABL tyrosine kinase inhibitor (TKI) imatinib: three studies demonstrated the involvement of the upregulation of the CXCR4/CXCL12 and IL-7 pathways [133,134,135]. It is noteworthy that through CXCL12-targeted deletion in the BM hematopoietic niche, MSCs are capable of promoting CML leukemic stem cell expansion despite TKI treatment [136].

In coculture with BM-MSCs, CLL cells display marked chemoresistance towards both classic chemotherapy and new biologic drugs. Reduced antileukemic fludarabine activity seems to be related to the upregulation of antiapoptotic Mcl-1 and Bcl-2 [137]. BM-MSCs also protected CLL cells from dexamethasone- and cyclophosphamide-induced apoptosis by maintaining Mcl-1 and protecting cells from PARP cleavage [138]. Stamatopoulos et al. demonstrated that antagonizing the SDF-1α/CXCR4 axis can increase drug-related apoptosis of CLL cells when in contact with BM-MSCs [139]. Interestingly, BM-MSCs can modulate the redox status of CLL cells, promoting cell survival and drug resistance. Additionally, EVs derived from BM-MSCs have been shown to increase the resistance of CLL cells to different drugs, including purine nucleoside analogs, corticosteroids, ibrutinib, idelalisib or venetoclax [140]. BM-MSCs also play a key role in the induction of drug resistance in MM cells [141]. The protective effect of BM-MSCs against bortezomib activity may be mediated by IL-8 secretion [142], miRNA transfer [143] or exosomes [144]. The production of fibroblast activation proteins (FAPs) by BM-MSCs has also been implicated in drug resistance via the activation of the β-catenin signaling pathway [145].

### 3.3. Impact of MSC Senescence in MM, CLL and Myelodysplastic Syndrome (MDS)

As described before, aging MSCs deeply modify many of their genetic and epigenetic activities, acquiring the so-called SASP, a peculiar condition characterized by increased production of proinflammatory molecules. Whether this phenotype promotes oncogenesis is still debated, as conflicting reports are present in the literature, suggesting a key regulatory role for cancer cells towards MSCs [35]. However, several reports have underlined how MSCs show a higher tendency to undergo senescence in some hematologic malignancy models, according to morphology and gene expression profile alterations.

Several studies have demonstrated that MM-MSCs display important differences compared to healthy donor MSCs (HD-MSCs): differences in gene and protein expression, functional alterations such as lower proliferation, decreased osteoblastic differentiation potential or impaired immunomodulation. Significant differences observed between BM-MSCs in an environment with or without the presence of MM cells demonstrated the ability of MM cells to modify BM-MSCs to increase their tumor-promoting effects [146]. However, some studies suggest that MM-MSCs are inherently abnormal, and these abnormalities remain even in the absence of MM cells.

Various studies have reported senescence-associated constitutive abnormalities in MM-MSCs leading to abnormal cell characteristics and increased tumor support. In a coculture model of MM cell lines and senescent MSCs, it has been described how cancer cell priming switches MSC secretory activity, reducing the production of prosenescent and apoptotic molecules and favoring angiogenesis and proliferation [147]. André et al. demonstrated that MM-MSCs display a SASP, lower proliferative rate, higher cell size, increased β-galactosidase activity and retention of cells in the S phase of the cell cycle [148]. This increased senescence was associated with reduced cyclin E1 and increased cyclin D1 expression as well as the overexpression of miRNAs related to aging compared to HD-MSCs [149]. In comparison to HD-MSCs, MM-MSCs express increased basal levels of IL-1β and TNF-α [150]. As reported by Zdzisinska et al., cytokines overexpressed by MM-MSCs can function as growth factors for MM cells and induce migration, adhesion, osteoclastogenesis and angiogenesis [151]. MM-MSCs exhibited long-term hematopoietic support and produced abnormally high amounts of IL-6 in the absence of any detectable MM cells [152]. IL-6 is a known protumoral cytokine favoring cancer cell formation and disease progression, which has also been associated with dexamethasone resistance and plasma cell retention in BM [150,153]. A recent study indicated that myeloma-associated elongation of telomere length of BM-MSCs may be a key element contributing to increased IL-6 expression, by which MSCs may facilitate MM development [154].

It has also been reported that the pathogenesis of bone lesions in this disease comes partly from senescent MSCs and their active impairment of osteoblast activity. Indeed, the fivefold higher expression of the osteoblast inhibitor DKK1, both at the transcript and protein levels, in MM-MSCs than in HD-MSCs suggests a direct role in osteolytic lesion propagation through autocrine and paracrine signaling [155]. The low rate of osteogenic differentiation is in part due to increased expression of inflammatory cytokines, such as TNF-α, able to suppress expression of TAZ, a Runx2/Cbfa1 transcriptional coactivator [156]. EphrinB2 and EphB4 expression in MM-MSCs was lower than in HD-MSCs and this dysregulated signaling may also decrease their osteogenic potential [157]. The MSC gene expression profile in MM patients seems to differ according to disease status after treatment: minimal residual disease (MRD)-negative patients express a completely different profile compared to the pretreatment phase. This observation suggests that MSC activity could play an important role in sustaining neoplastic proliferation, but a direct connection between the two elements has to be demonstrated [83]. Several gene expression profiling studies have extracted many genes differentially expressed by MM-MSCs and HD-MSCs. These genes are principally involved in tumor-microenvironment cross-talk, coding for proteins involved in MM cell growth, angiogenesis and osteoblast differentiation [158]. Furthermore, other significant differences involve important biological processes, such as the cell cycle, DNA repair, cell adhesion and metabolism [77,148]. In the study of Fernando et al., the downregulated genes were related to cell cycle progression, immune system activation and bone metabolism, suggesting that MM-MSCs might contribute to immune evasion and play a role in bone lesions [159]. Other genes related to different pathways of the immune system, including antigen processing and presentation, are altered in MM-MSCs compared to HD-MSCs [159]. MM-MSCs present reduced inhibitory efficiency towards T lymphocyte proliferation and reduced production of TGF-β, which could lead both to escape immune control and to reduced apoptosis of MM cells [152]. The studies evaluating the impact of MSCs in MM are summarized in Table 1.

In CLL, the relationship between MSCs and neoplastic cells has been widely documented, as discussed previously. CLL-MSCs display intrinsic qualitative and quantitative abnormalities that may be implicated in disease development and/or progression. The impaired proliferative potential of CLL-MSCs can be attributed, at least in part, to increased cell apoptosis. BM-MSCs from CLL patients seem to be less numerous, to present reduced proliferation potential and to express SASP, particularly characterized by increased production of IL-6, IL-8 and VEGF [160]. The documented abnormal production of CXCL12 and TGF-β from MSCs could represent a key mechanism for leukemic progression [161]. However, CLL-MSCs display normal immunosuppressive properties in terms of their capacity to suppress T-cell proliferative responses [161].

Most of these MSC features seem to be induced by the interaction with CLL cells, as demonstrated by a coculture experience reported by Ding and colleagues: a transcriptome analysis revealed an altered expression profile concerning genes mostly involved in senescence and cell cycle regulation, such as LIF, CDKN2B, DKK2, HGF, and FOXQ1 [162]. EVs also play a key role in this cross-talk mechanism among CLL cells and MSCs, as demonstrated in several studies [163]. In comparison to HD-MSCs, CLL-MSCs produced more EVs able to rescue CLL cells from apoptosis and induce higher migration activity and gene modifications than healthy Evs [140]. Moreover, CLL-derived exosomal proteins and miRNAs can induce an inflammatory phenotype in MSCs, enhancing the proliferation, migration and secretion of inflammatory cytokines [164]. All of the discussed studies on MSC aging and CLL are listed in Table 2.

Similar findings were reported in MDS, in which MSCs display reduced colony-forming and proliferation capacities and activation of the p53-p21 pathway, promoting the formation of a BM environment hostile towards normal hematopoiesis and finally favoring oncogenesis [165]. Alterations in cell cycle control have been found in MDS-MSCs: higher expression of cyclin-dependent kinase inhibitor 2B (CDKN2B) could be responsible for the low proliferative capacity of MSCs, favoring clonal progression [166]. MDS-MSCs also displayed a shift towards increased apoptosis, lower expression of VEGF, SCF and ANGPT, aberrant expression patterns of the Notch signaling pathway and increases in Wnt signaling inhibitors [167]. Among MSCs subpopulations, CD271+ MSCs are expanded in MDS and are in tight contact with HSCs in perivascular regions: these MSCs express abnormal levels of CXCL12, a chemokine promoting HSCs homing, and could be responsible for the abnormal localization of immature precursors (ALIP), a typical feature of the disease [101]. Senescent MSCs, through the increased production of cytokines such as IL-6, show the ability to stimulate HSC proliferation and differentiation, decreasing stemness capacity and promoting genome instability [51]. In addition, a recent review emphasizes the deleterious influence of an inflammatory environment on the selection of mutant HSCs carrying CHIP, with evident consequences for tumorigenesis [168]. The immunomodulatory capacity of MDS-MSCs is deeply modified under physiological conditions: the capacity of MDS-MSCs to inhibit T lymphocyte activation and proliferation is impaired in vitro [169]. Moreover, global activation of inflammatory patterns (NF-кB, EGF, TGF-β, and TNF signaling) and overexpression of negative regulators of hematopoiesis were described [170,171]. Epigenetic regulation, such as hypermethylation, seems to confer reduced growth capacity and osteogenic differentiation [172]. Hypomethylating agents, often used in high-risk MDS treatment, have been found to restore a normal MSC phenotype in patients achieving complete hematologic remission [173]. The level of expression of DICER-1 was lower in MSCs from MDS patients, altering their miRNA content [174]. Interestingly, some miRNAs were overexpressed in EVs derived from MDS-MSCs, such as miR-10a and miR-15a, which are involved in cell cycle proliferation and apoptosis and are able to modify hematopoietic cell properties [175]. Studies on the role of senescent activity in MDS pathogenesis are summarized in Table 3.

The supportive role exerted by the microenvironment, particularly MSCs, appears crucial in the pathogenesis and progression of certain hematologic malignancies, as was recently demonstrated in murine models of MDS [176]. The direct contribution of MSCs to leukemogenesis remains largely unknown. The characterization of “pathologic” MSCs requires in vitro expansion, which may alter their biological functions. In culture, MSCs undergo an aging process due to extensive proliferation. Several reports suggest that CLL-MSCs and MM-MSCs are probably dependent on leukemic clones for their long-term survival, as leukemic clones are dependent on MSCs for their own survival. Senescent MSCs in the BM of different hemopathies have never been directly demonstrated in vivo. In vivo tracking of MSC aging has been performed only in animal models and suggests a decline in MSC frequency with aging [177]. Age-related changes in BM-MSCs have been recently evaluated in uncultured BM-MSCs. A reduction in MSC colony number and density was confirmed in older donors, but multilineage gene expression profiles showed no age-related differences [178]. Recently, Alameda et al. [83] evaluated the impact of aging on healthy MSCs and determined how MM cells influence their functions: the transcriptional profile of “native” MSCs showed differences with the age of donors and in MM patients. Interestingly, fresh MM-MSCs are transcriptionally different from HD-MSCs and partially influenced by aging. The aging transcriptional changes are thus exacerbated by the cross-talk between MSCs and tumor cells. However, it has still not been clarified which actor in this cross-talk process is the first protein responsible for the pathway leading ultimately to hematopoietic niche alteration.

## 4. Conclusions

MSCs represent a key component of the BM microenvironment, exerting multiple functions that are fundamental for tissue homeostasis, such as the renewal of bone, adipose and connective tissues; the support of the hematopoietic niche; and the modulation of the immune system response. These activities are carried out through the secretion of a wide variety of compounds, such as growth factors, cytokines and EVs. The aging process determines profound modifications of both the morphology and functions of MSCs, of which the main modification is represented by the acquisition of the SASP, which strongly contributes to the development of a proinflammatory environment. Senescent MSCs play a key role in the development and progression of several solid tumors, and there is increasing evidence that they provide the inflammatory microenvironment supporting the progression of hematologic malignancies: indeed, MSCs reduce the apoptosis of cancer cells, induce chemoresistance and reduce the support of the hematopoietic niche, as comprehensively demonstrated in three oncohematologic models, MM, CLL and MDS. Whether MSC protumoral activity is the primum movens of clonal development or the effect of neoplastic stimulation still needs to be clarified. However, this activity seems to be of crucial importance for tumoral progression, opening the field for better comprehension of these diseases and potential therapeutic approaches.

## Figures and Tables

**Figure 1 cancers-13-00068-f001:**
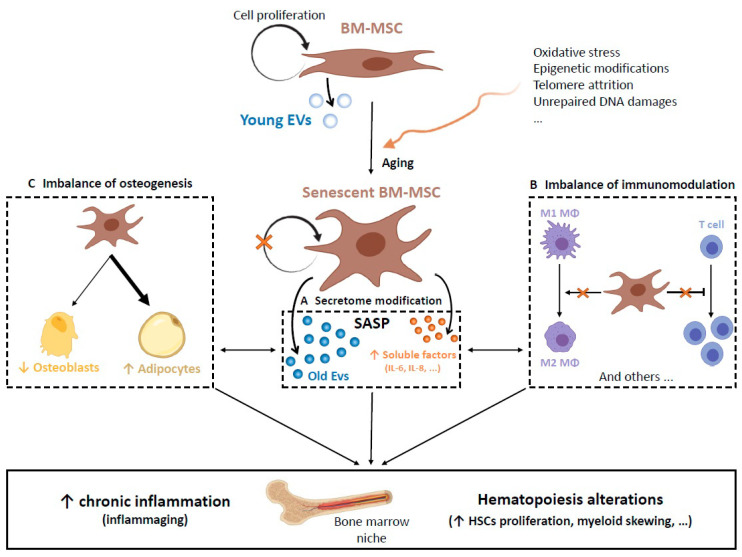
With aging, several factors induce the senescence of bone marrow mesenchymal stromal cells (BM-MSCs) that accumulate inside the bone marrow (BM) niche. (**A**) The senescent BM-MSCs adopt the senescent-associated secretory phenotype (SASP) enriched in particular with proinflammatory cytokines, and their extracellular vesicle (EV) morphology and content are strongly modified (see Section 2.1). (**B**) The immunomodulatory properties of aged BM-MSCs are impaired. For example, their ability to promote macrophage (MΦ) polarization into the M2 phenotype and their capacity to inhibit T lymphocyte proliferation are reduced (see Section 2.2). (**C**) An imbalance between osteogenesis and adipogenesis occurs, leading to a progressive replacement of bone by fat (see Section 2.3). All of these processes are closely interconnected and can lead to the establishment of low-grade chronic inflammation and hematopoiesis alterations (Figure created with BioRender).

**Table 1 cancers-13-00068-t001:** Studies evaluating the activity of MSCs in MM setting, with focus on MSCs senescence-like modifications.

MSCs Source	Coculture	Mechanisms and Reference
HD	ARH-77 cell lines	Secretome from MSCs showed impaired antitumor activity [147].
MM patients, MGUS patients	-	↑ SA-βGalA, cell size and hematopoietic support. ↓ proliferative capacity, osteoblastogenesis and immunomodulatory activity. Expression of senescence-associated secretory phenotype (SASP) [148].
MM patients	KMS12-PE cell lines	↑ SA-βGalA and overexpression of miR-485-5p and miR-519d. Cell cycle arrest in S phase. MM cells decreased SA-βGalA and influenced cell cycle characteristics of MM-MSCs [149].
MM patients	-	↑ basal levels of IL-1β and TNF-α [150].
MM patients	RPMI-8226 MM cell lines	↑ IL-6, IL-10, TNF-α, OPN, and especially HGF and BAFF production in response to MM cells. MSCs significantly enhanced the production of sIL-6R by MM cells [151].
MM patients, MGUS patients, Plasma cell leukemia patients	-	↑ IL-6 production. ↓ inhibitory capacity towards T lymphocyte proliferation. Characteristics also observed in the absence of any detectable tumor plasma cell [152].
MM patients	NCI-H929 MM cell lines	↑ SA-βGalA and tumor-supporting capacity. ↓ MSCs proliferation and differentiation potential. Dicer1 overexpression reversed the effects on differentiation and reduced cellular senescence. MM cells could induce the senescence of MSCs from HD. [153].
MM patients	-	↑ IL-6 and MIP-1α expression and telomere length. Telomere length is positively associated with the expressions of IL-6 and MIP-1α at the mRNA level in MM-MSCs [154].
MM patients	-	↑ DKK1 expression at transcript and protein levels [155].
MM patients	RPMI-8226 and U-266 cell lines	MM cells inhibited osteogenesis of MSCs from HD, which were associated to a reduced TAZ expression, partially restored by neutralization of TNF-α [156].
MM patients, SCID-Hu MM murine model	-	↓ levels of EFNB2 and EPHB4. EPHB4-Fc treatment inhibited MM growth, osteoclastosis, angiogenesis and stimulated osteoblastogenesis in vivo. EFNB2-Fc stimulated angiogenesis and osteoblastogenesis but had no effect on osteoclastogenesis and MM growth [157].
MM patients, MGUS patients	XG-1 and MOLP-6 MM cell lines	GDF15 induced dose-dependent growth of MM cells. ↓ MM-MSCs osteogenic differentiation capacity [158].
MM patients	-	Distinct gene expression profile between MM-MSCs and HD-MSCs (485 differentially expressed genes). In particular: ZNF521 and SEMA3A, involved in bone metabolism and, HLA-DRA and CHIRL1, implicated in the activation of immune response [159].
MM patients	-	Only 3 genes: DUSP2, MZB1, and TSPAN7, were significantly altered in MSCs isolated from MRD+ patients as compared to diagnosis. By contrast, 56 genes were significantly deregulated in MRD- MSCs compared to the time of diagnosis [83].

MSCs = mesenchymal stromal cells; MM = multiple myeloma; HD = healthy donors; MGUS = monoclonal gammopathy of uncertain significance; ↑ = increase or upregulation; SA-βGalA = senescence-associated β-galactosidase activity; ↓ = decrease or downregulation; SASP = senescence-associated secretory phenotype; MRD = minimal residual disease

**Table 2 cancers-13-00068-t002:** Studies evaluating the activity of MSCs in CLL setting, with focus on MSCs senescence-like modifications.

MSCs Source	Coculture	Mechanisms and Reference
CLL patients	-	↑ SA-βGalA. ↓ CFU-F and proliferative capacity. Polygonal aspect and expression of SASP [160].
CLL patients	B and T lymphocytes	CLL-MSCs presented impaired reserves, defective cellular growth and aberrant production of SDF-1 and TGF-β1, crucial cytokines for leukemic cells survival [161].
CLL patients	CLL patients’ cells	↑ of cycle inhibitors p16 and p57 expression, both key markers of cell senescence in CLL-MSCs. ↑ of Wnt inhibitors DKK1/DKK2 and Wnt5b expression in CLL-MSCs. MSCs co-culture with CLL cells induced altered expression of ~1500 genes mostly involved in regulation of cell growth and senescence (CDKN2B, DKK2, LIF, HGF, FOXQ1) and determined increased production of cytokines associated to SASP (MCP-1/IL-8/IL-6/IL-1Ra) [162].
CLL patients (MSCs-EVs)	CLL patients’ cells	MSC-EVs decreased apoptosis of CLL cells and increased chemoresistance towards several drugs, including fludarabine, ibrutinib, idelalisib and venetoclax. Enhanced both spontaneous and SDF-1α -induced migration capacities of CLL cells. Different gene expression profile between CLL cells cultured with or without EVs: overexpression of genes involved in the BCR pathway such as CCL3/4, EGR1/2/3, and MYC [140].
CLL patients (MSCs-EVs) MEC-1-eGFP CLL murine model	-	The transfer of CLL exosomal protein and microRNA induced an inflammatory phenotype in MSCs, determining increased proliferation, migration and secretion of inflammatory cytokines, contributing to a tumor-supportive microenvironment. Coinjection of CLL-derived exosomes and CLL cells promoted tumor growth in immunodeficient mice [164].

MSCs = mesenchymal stromal cells; CLL = chronic lymphocytic leukemia; ↑ = increase or upregulation; SA-βGalA = senescence-associated β-galactosidase activity; ↓ = decrease or downregulation; CFU-F = colony-forming unit-fibroblast; SASP = senescence-associated secretory phenotype; BCR = B-cell receptor; MSC-EVs = MSCs-derived extracellular vesicles

**Table 3 cancers-13-00068-t003:** Studies evaluating the activity of MSCs in MDS, with focus on MSCs senescence-like modifications.

MSCs Source	Coculture	Mechanisms and Reference
MDS patients	-	↑ cell size, SA-βGalA and p53 and p21 expression. ↓ proliferative capacity, colony-forming potential and hematopoietic supporting function. Alteration of cytoskeleton. Osteogenic differentiation potential of MDS-MSCs from lower risk MDS was impaired [165].
MDS patients	-	↑ CDKN2B expression 8–11 times higher in MDS-MSCs compared to HD-MSCs. ↓ proliferative capacity [166].
MDS patients	-	↑ apoptosis and Wnt signaling inhibitory ligands Dkk-1 and Dkk-2 expression. ↓ VEGF, SCF and ANGPT expression with no change in the expression of CXCL12A and LIF. Significantly altered cell cycle status and aberrant expression pattern of Notch signaling components [167].
MDS patients	-	↓ hematopoietic cytokine expression. ↓ of the capacity of MDS-MSCs to inhibit T lymphocyte activation and proliferation in vitro [169].
MDS patients	HSPCs	Functional activation of NF-κB pathway in MDS-MSCs, resulting in impaired proliferation of MSCs, contributing to the reduced support for HSPCs in vitro [170].
MDS patients	-	↑ EGF and TGF-β expression. ↑ TNF signaling [171].
MDS patients	-	↓ osteogenic differentiation. Altered expression of key molecules involved in HSPCs supportive function, in particular osteopontin, Jagged1, Kit-ligand and Angiopoietin [172].
MDS patients	-	Hypomethylating agents restore a normal MSC phenotype in patients achieving hematologic complete remission [173].
MDS patients	-	↓ DICER1 expression. ↓ mir-155, miR-181a and miR-222 expression [174].
MDS patients (MSCs-EVs)		Some microRNAs were overexpressed in MSCs-EVs and two of them, miR-10a and miR-15a, were confirmed by polymerase chain reaction. If transferred to CD34+ cells, these microRNAs modify the expression of MDM2 and P53 genes. Higher cell viability and clonogenic capacity after MSCs-EVs inclusion in CD34+ cells [175].

MSCs = mesenchymal stromal cells; MDS = myelodysplastic syndrome; HD = healthy donors; ↑ = increase or upregulation; SA-βGalA = senescence-associated β-galactosidase activity; ↓ = decrease or downregulation; HSPCs = hematopoietic stem and progenitor cells; MSC-EVs = MSCs-derived extracellular vesicles

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
