# Peer review of "Aging of Bone Marrow Mesenchymal Stromal Cells: Hematopoiesis Disturbances and Potential Role in the Development of Hematologic Cancers"

_cancers, 2020, doi:10.3390/cancers13010068_

Round 1

Reviewer 1 Report

Review article "Aging of Bone Marrow Mesenchymal Stromal Cells: Hematopoiesis Disturbances and Potential Role in the Development of Hematologic Cancers"

Multiple intercellular interactions and bidirectional crosstalk are tightly regulated in the bone marrow to maintain the stem cell capacity. Mesenchymal stromal cells (MSCs) are an important cellular component of the bone marrow microenvironment. They are multipotent and can actively support hematopoietic stem cells (HSCs) but also leukemic cells. Therefore, MSCs may act differently under diverse conditions. Since aging is potentially coupled to an increased risk for the development of malignant disorders the understanding of underlying mechanisms is important also for the development of therapeutic strategies.

In this manuscript Massaro et al. rewiewed main features of MSC senescence and implication in hematologic cancer progression. A main concern is that a lot of other review articles are cited but primary publications are missing.

Other specific comments:

Page 2

Line 44: disruption? Remodeling would be the better term

Line 73: destabilization?, "susceptibility" instead of "sensitivity"

Line 76-80: references?

Line 77: Chronic inflammation is the result of senescent-associated secretory phenotype (SASP) as well as of the myeloid bias. This needs to be explained more precise.

Page 3

Figure 1: Old EVs are part of the SASP. The figure content is poorly explained in the text and references for the hypotheses are missing.

Line 93: time-limited? Could be rather a limitation in a certain number of divisions.

Line 98-104: The paragraph of EVs is improperly placed since the SASP is discussed later.

Line 100: "causing aging" is too general, causing functional impairment of HSCs would be better

Line 101: The EV secretion is increased but not more numerous.

Line 108: It is not clear why aging of MSCs should cause mutations in epigenetic regulators in HSCs. The impact of epigenetic changes in MSCs is not discussed.

Page 4

Line 120: It is difficult to discuss investigations of the SASP without an introduction of replicative senescence and radiation-induced aging as model, respectively, since the most studies used long-term cultured and passaged young MSCs which can only be limited translated to physiological aging. Research articles are not cited, only reviews were included. At least the paper of Carlos Sepúlveda et al. (Stem Cells, 2014) should be mentioned.

Line 122: Provide the original citation when distinct data such as cytokine levels are presented.

Line 125: Ref 37 confers to dermal fibroblasts, not to MSCs.

Page 5

Line 173: The differences between old and young EVs  might be (not "are") due to their miRNA content.

Line 187-200: Wagner et al. (PLoS One, 2008) have shown that replicative senenscent MSCs behave in the opposite way, with a higer osteogenic differentiation potential.

Line 202: Reference?

In general, in this paragraph the association between MSCs and hematopoiesis is missing.

Walenda et al (Journal of Cellular and Molecular Medicine, 2010) and Kulkarni et al. (Stem Cells, 2018) have published relevant studies with MSCs and MSC-EVs, respectively.

Page 7

Line 279: MSCs modify many of (not "all") their genetic and epigenetic activities...

Page 8

Line 317: than in HD-MSC

Line 320 and following: The conclusion is not clear. The patients receive chemotherapy before autologous transplantation, which massively harm all blood cells, including myeloma cells but also the stroma. Genetic alterations after chemotherapy are not the proof that myeloma cells use the stromal microenvironment for maintenance or that stromal cells are not supportive anymore. For this conclusion in-vitro and in-vivo experiments are required.

Page 13

Line 403: disruption?

Author Response

Response to Reviewer 1 Comments
Dear Reviewer,
Many thanks for your reviewing and your constructive comments. Please find below the point-by-point response.
Kind regards,

Point 1: A main concern is that a lot of other review articles are cited but primary publications are missing

Response 1: we have added to our manuscript some primary and experimental publications. We have also reworked some paragraphs in order to be more balanced and exhaustive and we have referred to more research publications. Specifically, we have further detailed MSC senescence and MSC epigenetic modifications occurring with aging (see section 2.1) and the imbalance between osteogenesis and adipogenesis (see section 2.2). Below, we have indicated some of the primary publications added to our manuscript concerning:

- The support of HSCs by bone marrow niche and MSCs:
Taichman, R.S.; Emerson, S.G. Human osteoblasts support hematopoiesis through the production of granulocyte colony-stimulating factor. The Journal of experimental medicine 1994, 179, 1677-1682, doi:10.1084/jem.179.5.1677
Zhang, J.; Niu, C.; Ye, L.; Huang, H.; He, X.; Tong, W.G.; Ross, J.; Haug, J.; Johnson, T.; Feng, J.Q., et al. Identification of the haematopoietic stem cell niche and control of the niche size. Nature 2003, 425, 836-841, doi:10.1038/nature02041
Majumdar, M.K.; Thiede, M.A.; Haynesworth, S.E.; Bruder, S.P.; Gerson, S.L. Human marrow-derived mesenchymal stem cells (MSCs) express hematopoietic cytokines and support long-term hematopoiesis when differentiated toward stromal and osteogenic lineages. Journal of hematotherapy & stem cell research 2000, 9, 841-848, doi:10.1089/152581600750062264.
- The differentiation potential of MSCs:
Pittenger, M.F.; Mackay, A.M.; Beck, S.C.; Jaiswal, R.K.; Douglas, R.; Mosca, J.D.; Moorman, M.A.; Simonetti, D.W.; Craig, S.; Marshak, D.R. Multilineage potential of adult human mesenchymal stem cells. Science (New York, N.Y.) 1999, 284, 143-147, doi:10.1126/science.284.5411.143.
- The immunomodulation properties of MSCs:
Waterman, R.S.; Tomchuck, S.L.; Henkle, S.L.; Betancourt, A.M. A new mesenchymal stem cell (MSC) paradigm: polarization into a pro-inflammatory MSC1 or an Immunosuppressive MSC2 phenotype. PloS one 2010, 5, e10088, doi:10.1371/journal.pone.0010088.
Kim, J.; Hematti, P. Mesenchymal stem cell-educated macrophages: a novel type of alternatively activated macrophages. Experimental hematology 2009, 37, 1445-1453, doi:10.1016/j.exphem.2009.09.004.
- Cell senescence, MSC senescence and MSC senescence-induced conditions:
Hayflick, L. THE LIMITED IN VITRO LIFETIME OF HUMAN DIPLOID CELL STRAINS. Experimentalcell research 1965, 37, 614-636, doi:10.1016/0014-4827(65)90211-9.
Baxter, M.A.; Wynn, R.F.; Jowitt, S.N.; Wraith, J.E.; Fairbairn, L.J.; Bellantuono, I. Study of telomere length reveals rapid aging of human marrow stromal cells following in vitro expansion. Stem cells (Dayton, Ohio) 2004, 22, 675-682, doi:10.1634/stemcells.22-5-675.
Cmielova, J.; Havelek, R.; Soukup, T.; Jiroutová, A.; Visek, B.; Suchánek, J.; Vavrova, J.; Mokry, J.; Muthna, D.; Bruckova, L., et al. Gamma radiation induces senescence in human adult mesenchymal stem cells from bone marrow and periodontal ligaments. International journal of radiation biology 2012, 88, 393-404, doi:10.3109/09553002.2012.666001.
Zhou, S.; Greenberger, J.S.; Epperly, M.W.; Goff, J.P.; Adler, C.; Leboff, M.S.; Glowacki, J. Age-related intrinsic changes in human bone-marrow-derived mesenchymal stem cells and their differentiation to osteoblasts. Aging cell 2008, 7, 335-343, doi:10.1111/j.1474-9726.2008.00377.x.
- MSC epigenetic modifications and aging:
Bork, S.; Pfister, S.; Witt, H.; Horn, P.; Korn, B.; Ho, A.D.; Wagner, W. DNA methylation pattern changes upon long-term culture and aging of human mesenchymal stromal cells. Aging cell 2010, 9, 54-63, doi:10.1111/j.1474-9726.2009.00535.x.

Point 2:
Line 44: disruption? Remodeling would be the better term
Response 2: we agree and made the change:
“A growing number of studies point to a clear link between aging, remodeling of the BM microenvironment and impairment of hematopoiesis, leading, among other things, to hematologic cancers”
Point 3: Line 73: destabilization?, "susceptibility" instead of "sensitivity"
Response 3: we agree and made the change:
“The result is an alteration of the immune system, leading to an increased susceptibility to infections and to the development of autoimmune diseases and cancers”
Point 4: Line 76-80: references?
Response 4: we have modified this section to mention only the extrinsic factors related to BM-MSCs which are detailed below in the review.
Point 5: Line 77: Chronic inflammation is the result of senescent-associated secretory phenotype (SASP) as well as of the myeloid bias. This needs to be explained more precise.
Response 5: we agree with you and we are well aware that the chronic inflammation state observed during aging is not only due to the compounds secreted by cells adopting SASP. For our manuscript, we have decided to focus only on direct consequences of BM-MSC aging on the inflammatory state of the bone marrow and hematopoiesis, but not on the subsequent consequences leading to chronic inflammation in the body. However, it is clear that hematopoiesis disturbances and myeloid bias, which are not only due to BM-MSC aging, have an additional impact on inflammaging. For example, it has been shown that aging is accompanied by an increase in generation of myeloid-derived suppressor cells (MDSCs) and these cells appear to be important for the establishment and maintain of inflammaging. This issue is well addressed in the review made by Salminen et al. (Ageing Res Rev, 2018). This is an interesting area of research but we feel it is beyond the scope of our article and we have decided not to discuss it further.
Point 6: Figure 1: Old EVs are part of the SASP. The figure content is poorly explained in the text and references for the hypotheses are missing.
Response 6: we have adapted the figure for including Old EVs in SAPS. With this figure we want to provide a simple overview of the three mains mechanisms described in our manuscript by which aging of BM-MSCs may contribute to hematopoiesis disturbances. Each one is delimited by a specific area on the Figure (with a dotted rectangle) and we added in the description the section number in which it is explained in more detail with the appropriate references.

Point 7: Line 93: time-limited? Could be rather a limitation in a certain number of divisions.
Response 7: we agree and made the change:
“they can only undergo a limited number of cell divisions before entering a senescent state”.
Point 8: Line 98-104: The paragraph of EVs is improperly placed since the SASP is discussed later.
Response 8: indeed you are right. We have rearranged this paragraph and moved further down in the paragraph concerning the SASP:
“The alterations associated with BM-MSC senescence also lead to a deep modification of their secretome, making them adopt a new phenotype called the senescent-associated secretory phenotype (SASP)[35]. This SASP is characterized by increased secretion of growth factors, proangiogenic factors, extracellular matrix remodeling factors and especially proinflammatory cytokines such as IL-1β, IL-6 and IL-8[48-52]. It is now well known that EVs contribute greatly to the SASP and that senescence of MSCs has a strong impact on them: their secretion is increased while their size is reduced and their content is modified, especially in terms of miRNA[18,53]. For example, the activation of AKT in aged BM-MSCs leads to increased partitioning of miR-17 and miR-34a in EVs, which, upon transfer to HSCs, cause functional impairment via downregulation of autophagy-related genes[54]. Terlecki-Zaniewicz and colleagues suggested that EVs of dermal fibroblasts and their miRNAs act as cargo for novel members of the SASP that are selectively secreted or retained in cellular senescence[55]. Although there are no similar experimental data on MSCs, it is reasonable to assume that this may be applicable also to them. Robbins suggested that senescent cell-derived EVs could function as pro-geronic factors[56]. The SASP participates in the establishment of the low-grade and chronic inflammation state observed during aging, called inflammaging[57,58].”
Point 9: Line 100: "causing aging" is too general, causing functional impairment of HSCs would be better
Response 9: we agree and made the change:
“the activation of AKT in aged BM-MSCs leads to increased partitioning of miR-17 and miR-34a in EVs, which, upon transfer to HSCs, cause functional impairment via downregulation of autophagy-related genes”
Point 10: Line 101: The EV secretion is increased but not more numerous.
Response 10: we agree and made the change:
“senescence of MSCs has a strong impact on them: their secretion is increased while their size is reduced and their content is modified, especially in terms of miRNA”
Point 11: Line 108: It is not clear why aging of MSCs should cause mutations in epigenetic regulators in HSCs. The impact of epigenetic changes in MSCs is not discussed.
Response 11: in order to address this question, we have added a paragraph talking about epigenetic modifications associated to MSC aging:
“Epigenetic modifications are key components of the BM niche homeostasis and can contribute to age- and disease-associated MSC alterations. Modifications of MSC DNA methylation patterns and hypermethylated and hypomethylated CpG sites in several genomic loci have been observed upon aging and replicative senescence[37,38]. Some epigenetic regulators have been identified to participate in MSCs aging. The expression and activation of Sirt1, a NAD-dependent histone deacetylase, decrease with age and modify MSC proliferation and differentiation[39,40]. Interestingly, miR-199b-5p, which is predicted to target Sirt1, is deregulated in old BM-MSCs[41]. MSCs deficient in Sirt6, another histone deacetylase, displayed accelerated cellular senescence, dysregulated redox metabolism and increased sensitivity to oxidative stress[42]. In HSCs, identification of somatic mutations in the epigenetic regulators DNMT3, TET2 and ASXL1 is associated with an increased risk of developing hematologic cancers[43]. These mutations can occur as people age and their identification in healthy people is known as clonal hematopoiesis of indeterminate potential (CHIP). These three epigenetic regulators seem to be involved in MSC aging. In human umbilical cord blood-derived MSCs (UC-MSCs), the inhibition of DNMT1 and DNMT3b induces cellular senescence[44]. In a mouse model used to study age-related skeletal diseases, the expression of TET2 resulted decreased[45]. In addition, it has been shown in mice that a loss of ASXL1 or TET2 impairs BM-MSCs fate and their ability to support hematopoiesis[46,47]. Taken together, these two observations suggest that epigenetic modifications of BM-MSCs occurring during aging can contribute to hematopoiesis disturbances.”
Point 12: Line 120: It is difficult to discuss investigations of the SASP without an introduction of replicative senescence and radiation-induced aging as model, respectively, since the most studies used long-term cultured and passaged young MSCs which can only be limited translated to physiological aging. Research articles are not cited, only reviews were included. At least the paper of Carlos Sepúlveda et al. (Stem Cells, 2014) should be mentioned.
Response 12: we have added a paragraph talking about MSC senescence at the beginning of the section in which we describe SASP. We added more information concerning in vitro senescence and physiological aging with reference to research article:
“MSCs are multipotent cells with proliferative properties. However, similar to any normal cell, they can only undergo a limited number of cell divisions before entering a senescent state. Cellular senescence and its related cell cycle arrest were observed for the first time by Hayflick in long-term in vitro culture of human fibroblasts[25]. Since then, a wide variety of factors causing MSC senescence have also been described, such as oxidative stress[26], telomere attrition occurring during in vitro expansion[27] or unrepaired DNA damages[28]. Accumulation of senescent cells was also observed in several aged tissues, as it was well illustrated in a recent study which evaluated the expression of p16 and p21, two markers of senescence, in organs from young or old donors [29]. An increased level of p21 was also observed in BM-MSCs from elderly people, suggesting that senescent BM-MSCs accumulate with physiological aging[30]. Nevertheless, some experiments studying MSC senescence do not use cells form elderly donors but rather in vitro stress-induced-senescence conditions such as long-term culture expansion or senescence induced by gamma irradiation. It is therefore necessary to remain cautious when comparing data concerning in vitro senescence with physiological aging.”
We have also referred to the paper of Carlos Sepulveda in the section talking about the immunomodulation properties of MSCs:
“it has been shown that gamma-irradiated senescent BM-MSCs showed a lower capacity to migrate in response to proinflammatory signals and, at least in part, a lower inhibitory capacity towards T lymphocytes”
Point 13: Line 122: Provide the original citation when distinct data such as cytokine levels are presented.
Response 13: we have added two citations of research papers showing an increase of IL-1β, IL-6 or IL-8 in aged BM-MSCs:
O'Hagan-Wong, K.; Nadeau, S.; Carrier-Leclerc, A.; Apablaza, F.; Hamdy, R.; Shum-Tim, D.; Rodier, F.; Colmegna, I. Increased IL-6 secretion by aged human mesenchymal stromal cells disrupts hematopoietic stem and progenitor cells' homeostasis. Oncotarget 2016, 7, 13285-13296, doi:10.18632/oncotarget.7690.
Gnani, D.; Crippa, S.; Della Volpe, L.; Rossella, V.; Conti, A.; Lettera, E.; Rivis, S.; Ometti, M.; Fraschini, G.; Bernardo, M.E., et al. An early-senescence state in aged mesenchymal stromal cells contributes to hematopoietic stem and progenitor cell clonogenic impairment through the activation of a pro-inflammatory program. Aging cell 2019, 18, e12933, doi:10.1111/acel.12933.
Concerning IFN-γ, it seems indeed that no research article has shown its upregulation in the specific case of senescent BM-MSCs. This is a mistake on our part and we have deleted its mention.
Point 14: Line 125: Ref 37 confers to dermal fibroblasts, not to MSCs.
Response 14: you are right, the article of Terlecki-Zaniewicz and colleagues refers to dermal fibroblasts. However, we think it is reasonable to assume that their observations could apply to MSCs and we have mentioned this:
“Terlecki-Zaniewicz and colleagues suggested that EVs of dermal fibroblasts and their miRNAs act as cargo for novel members of the SASP that are selectively secreted or retained in cellular senescence[55]. Although there are no similar experimental data on MSCs, it is reasonable to assume that this may also be applicable to them.”
Point 15: Line 173: The differences between old and young EVs might be (not "are") due to their miRNA content.
Response 15: we agree and made the change:
“These differences between old and young EVs might be due to their miRNA content.”
Point 16 and 17: Line 187-200: Wagner et al. (PLoS One, 2008) have shown that replicative senescent MSCs behave in the opposite way, with a higher osteogenic differentiation potential.
In general, in this paragraph the association between MSCs and hematopoiesis is missing.
Response 16 and 17: it is an interesting comment. We have decided to modify this section profoundly in order to mention the paper of Wagner et al. and to be more nuanced in the role of BM-MSCs in the imbalance between osteogenesis and adipogenesis. We have also added several references to research articles to be more complete concerning the consequences on hematopoiesis:
“Bone tissue is a dynamic tissue undergoing constant remodeling throughout its lifetime. Its homeostasis is maintained by two complementary processes: the formation of new bone by osteoblasts and the resorption of old and damaged tissues by osteoclasts. BM-MSCs play an important role in this balance by being recruited at the bone-resorptive site through TGF-β1 signaling and by differentiating into osteoblasts[75]. However, during aging, bone resorption increases, and the bone density of the organism progressively decreases, leading to osteoporosis and increasing the risk of fractures[76]. The age-related changes in MSC differentiation potential have been studied by several groups in mice and humans. Although conflicting results have been reported, one cause for the imbalance between bone and adipose tissue occurring with aging could be due to a gradual loss of the ability of BM-MSCs to differentiate into osteoblasts, favoring differentiation into adipocytes. A study using senescent BM-MSCs obtained after long-term culture showed an increased osteogenic differentiation potential after several passages[77]. However, other studies comparing BM-MSCs harvested from young and old donors have shown both a maintenance[78,79] or a decrease in the osteogenic differentiation of oldest BM-MSCs[30,80-82]. In a recent study, authors analyzed the transcriptional profile of freshly isolated BM-MSCs from young and old donors and showed the upregulation of genes implicated in the peroxisome proliferator-activated receptor (PPAR) signaling in the oldest group, suggesting a reinforcement of pro-adipogenic microenvironment with aging[83].
Several factors are implicated in the control of BM-MSC differentiation: RUNX2 and SP7 promote osteogenesis[84,85], while CEBPα, CEBPβ, CEBPγ and PPARγ promote adipogenesis[86]. There is a growing body of data highlighting the age-dependent control of these factors. In mice, it has been shown that FOXP1, a transcription factor interacting with CEBPβ, is downregulated during aging[87]. Similarly, CBFβ and MAF, two cofactors of RUNX2, are also downregulated with increasing age[88,89], while PPARγ is upregulated[90]. All of these signaling pathway modifications promote adipogenesis. The miRNA content of BM-MSCs and their EVs also seems to be involved in the imbalance between osteogenesis and adipogenesis. Indeed, it has been shown that aging and oxidative stress can alter the miRNA cargo of EVs, which in turn causes the suppression of cellular proliferation and osteogenic differentiation of BM stromal cells[91]. It has also been reported that miR-31a-5p level rises in aged BM-MSCs and appears to be involved in increasing adipogenesis and decreasing osteogenesis[92]. The decrease in osteogenic differentiation by BM-MSCs is accompanied by a reduced level of osteopontin secretion, which is known to negatively regulate the self-renewal of HSCs[93,94].
Adipocytes in BM impair hematopoiesis by diminishing the differentiation of hematopoietic progenitors towards the B lymphocyte lineage[95]. In a recent paper, Aguilar-Navarro et al. observed an increase of adipocytes in BM of elderly people associated with an increase of maturing myeloid cells and they proposed a contributive role for adipocytes in myeloid skewing[96]. Another study conducted on mice has shown that aging is associated with the expansion of adipogenic potential of a stem cell-like subpopulation in the BM which, in turn, altered hematopoiesis through an excessive production of Dipeptidyl peptidase-4[97].”
Point 18: Line 202: Reference?
Response 18: we have slightly modified this paragraph to be more nuanced and we have added a reference. The role of adipose tissue in inflammation occurring during aging is a broad research domain and many research articles studying it. As this is not the subject of our manuscript and it would be impossible to be exhaustive, we have preferred to mention a review as reference.
“By increasing the number of adipocytes inside the BM, the aging of BM-MSCs could also indirectly impacts the inflammatory state of the BM niche. It is indeed well known today that adipose tissue participates in the production of a large amount of soluble factors and cytokines and that aging and metabolic diseases, like obesity, are correlated with an increase of its proinflammatory cytokine secretion[98].”
Mau, T.; Yung, R. Adipose tissue inflammation in aging. Experimental gerontology 2018, 105, 27-31, doi:10.1016/j.exger.2017.10.014.
Point 19: Line 279: MSCs modify many of (not "all") their genetic and epigenetic activities...
Response 19: we agree and made the change:
“As described before, aging MSCs deeply modify many of their genetic and epigenetic activities … ”
Point 20: Line 317: than in HD-MSC
Response 20: we agree and made the change:
“EphrinB2 and EphB4 expression in MM-MSCs was lower than in HD-MSCs and this dysregulated signaling may also decrease their osteogenic potential.”
Point 21: Line 320 and following: The conclusion is not clear. The patients receive chemotherapy before autologous transplantation, which massively harm all blood cells, including myeloma cells but also the stroma. Genetic alterations after chemotherapy are not the proof that myeloma cells use the stromal microenvironment for maintenance or that stromal cells are not supportive anymore. For this conclusion in-vitro and in-vivo experiments are required.
Response 21: we agree this is not the proof and we changed the statement in the paper. However the two groups (MRD+ vs MRD-) were equally exposed to treatment so it is difficult to say that the two different gene expression profiles are related only to chemotherapy effect.
Point 22: Line 403: disruption?
Response 22: Response 20: we agree and made the change:
“However, it has still not been clarified which actor in this cross-talk process is the first protein responsible for the pathway leading ultimately to hematopoietic niche alteration.”
Point 23: Extensive editing of English language and style required
Response 23: we are not native English speakers and we submitted our manuscript to AJE (https://www.aje.com/) for the editing prior to submission to MDPI. You can see the certificate of editing below.(please see attachment)

Reviewer 2 Report

This is an extremely well written and informative review of the subject.  The only concern that this reviewer has is with the statement that MSC self-renew.  Can the author's provide experimental evidence that this is the case?  Otherwise, please just simply state that MSCs proliferate with multi-potential capability.

Author Response

Response to Reviewer 2 Comments
Dear Reviewer,
Many thanks for your reviewing and your comment. You are right concerning the self-renewing capacities of MSCs and we have made modification in our manuscript :
Line 103: “MSCs are multipotent cells with proliferative properties.”
Kind regards,

Reviewer 3 Report

This review is well written and comprises a nice overview on the role of MSC in aging bone marrow and the subsequent consequences for the development and progression of hematologic cancers.

Major comment:

While there is a specific paragraph on the role of BM-MSC during aging, there is one aspect barely discussed involving the altering composition of the MSC compartment during aging. There is enough literature pointing towards this aspect (see some suggestions below) and the supposed altered composition in terms of MSC subsets might also explain the SASP phenotype that is described by the authors. Therefore, I would suggest to supply this review with a chapter on this aspect including at least the references mentioned below:

Another aspect that might be mentioned or at least briefly discussed: the role of MSC in a metastatic context: e.g. what is the role of MSC in solid tumors like neuroblastoma or breast cancer that metastasize to the bone marrow microenvironment?

Literature:

Haematopoietic stem cell activity and interactions with the niche. Pinho S, Frenette PS.Pinho S, et al. Among authors: frenette ps. Nat Rev Mol Cell Biol. 2019 May;20(5):303-320. doi: 10.1038/s41580-019-0103-9.

Low/negative expression of PDGFR-α identifies the candidate primary mesenchymal stromal cells in adult human bone marrow.Li H, Ghazanfari R, Zacharaki D, Ditzel N, Isern J, Ekblom M, Méndez-Ferrer S, Kassem M, Scheding S.Li H, et al. Among authors: mendez ferrer s. Stem Cell Reports. 2014 Dec 9;3(6):965-74. doi: 10.1016/j.stemcr.2014.09.018. Epub 2014 Oct 30.

The composition of the mesenchymal stromal cell compartment in human bone marrow changes during development and aging. Maijenburg MW, Kleijer M, Vermeul K, Mul EP, van Alphen FP, van der Schoot CE, Voermans C.Maijenburg MW, et al. Haematologica. 2012 Feb;97(2):179-83. doi: 10.3324/haematol.2011.047753. Epub 2011 Oct 11.

Distinctive contact between CD34+ hematopoietic progenitors and CXCL12+ CD271+ mesenchymal stromal cells in benign and myelodysplastic bone marrow. Flores-Figueroa E, Varma S, Montgomery K, Greenberg PL, Gratzinger D.Flores-Figueroa E, et al. Lab Invest. 2012 Sep;92(9):1330-41. doi: 10.1038/labinvest.2012.93. Epub 2012 Jun 18.

CD271 antigen defines a subset of multipotent stromal cells with immunosuppressive and lymphohematopoietic engraftment-promoting properties. Kuçi S, Kuçi Z, Kreyenberg H, Deak E, Pütsch K, Huenecke S, Amara C, Koller S, Rettinger E, Grez M, Koehl U, Latifi-Pupovci H, Henschler R, Tonn T, von Laer D, Klingebiel T, Bader P.Kuçi S, et al. Haematologica. 2010 Apr;95(4):651-9. doi: 10.3324/haematol.2009.015065. Epub 2010 Feb 23.

CD146 expression on primary nonhematopoietic bone marrow stem cells is correlated with in situ localization. Tormin A, Li O, Brune JC, Walsh S, Schütz B, Ehinger M, Ditzel N, Kassem M, Scheding S.Tormin A, et al. Blood. 2011 May 12;117(19):5067-77. doi: 10.1182/blood-2010-08-304287. Epub 2011 Mar 17.

The neural crest is a source of mesenchymal stem cells with specialized hematopoietic stem cell niche function. Isern J, García-García A, Martín AM, Arranz L, Martín-Pérez D, Torroja C, Sánchez-Cabo F, Méndez-Ferrer S.Isern J, et al. Elife. 2014 Sep 25;3:e03696. doi: 10.7554/eLife.03696.

A Subpopulation of Stromal Cells Controls Cancer Cell Homing to the Bone Marrow. Rossnagl S, Ghura H, Groth C, Altrock E, Jakob F, Schott S, Wimberger P, Link T, Kuhlmann JD, Stenzl A, Hennenlotter J, Todenhöfer T, Rojewski M, Bieback K, Nakchbandi IA.Rossnagl S, et al. Cancer Res. 2018 Jan 1;78(1):129-142. doi: 10.1158/0008-5472.CAN-16-3507. Epub 2017 Oct 24.

Author Response

Response to Reviewer 3 Comments
Dear Reviewer,
Many thanks for your reviewing and your constructive comments. Please find below the point-by-point response.
Kind regards,

Point 1: While there is a specific paragraph on the role of BM-MSC during aging, there is one aspect barely discussed involving the altering composition of the MSC compartment during aging. There is enough literature pointing towards this aspect (see some suggestions below) and the supposed altered composition in terms of MSC subsets might also explain the SASP phenotype that is described by the authors. Therefore, I would suggest to supply this review with a chapter on this aspect including at least the references mentioned below:

Response 1: we agree with your suggestion and we have added a specific paragraph about the functional and spatial heterogeneity of BM-MSCs:

2.4. Functional and spatial heterogeneity of BM-MSCs
MSCs represent a complex cell population characterised by specific localisation and functional heterogeneity that may be essential to their biological role. Several surface markers can be used to identify the different subpopulations of MSCs[1]. In BM, CD271 antigen can be used to identify a subset of BM-MSCs able to inhibit the proliferation of allogenic T-lymphocytes and presenting lympho-hematopoietic engraftment-promoting properties[2]. Most of HSCs are located in intimate cell-cell contact with these CD271+ MSCs[3]. A low or negative expression of platelet derived growth factor receptor alpha (PDGRF-α) by CD271+ MSCs is correlated with expression of key-genes for HSC supportive function and this expression is modulated according to the different phases of development of the organism[4]. CD271+ cells can be further divided in two cell subgroups with different localisation depending on the expression of CD146. CD146+ status defines MSC population located in the perivascular spaces while CD146- cells are found in the endosteal region[5]. These populations have different degree of maturity: CD146- MSCs are more mature whereas CD146+ cells retain plasticity. Their distribution varies with age: Maijenburg et al. showed a predominance of CD146+ subset in pediatric and fetal BM and suggested that variation in MSC subpopulations is a dynamic process that can change MSC functions during aging of the BM [6].
Other studies using a new method of single cell transcriptional analysis showed age-related changes in BM-MSCs composition. Duscher et al. identified an age-related depletion of a subpopulation characterized by a pro-vascular transcriptional profile[7]. More recently, Khong and colleagues identified a unique quiescent subpopulation exclusively present in MSCs from young donors and showed that this subpopulation was characterized by a higher expression of genes involved in tissue regeneration[8].
It has also been described the existence of two populations of MSCs with neural crest or mesoderm embryonic origins and particularly the neural crest has been proposed as a source of MSCs with specialized hematopoietic stem cell niche function[9]. Embryonic origin has also been shown to play an essential role in the age-related decrease in the functional capacities of BM-MSCs[10].

Point 2: Another aspect that might be mentioned or at least briefly discussed: the role of MSC in a metastatic context: e.g. what is the role of MSC in solid tumors like neuroblastoma or breast cancer that metastasize to the bone marrow microenvironment?

Response 2: we decided to focus on our manuscript on hematologic malignancies but, indeed, there are some interesting research articles showing connexions between MSCs and metastatic process in solid tumors. Thus, we have added a short paragraph at the beginning of the section “BM-MSCs and hematologic malignancies” to mentioned it:

BM-MSCs play a dual role in tumor cell growth in vitro and in vivo: they suppress tumor cell proliferation and inhibit tumor growth, but they also suppress tumor cell apoptosis and promote tumor growth[11,12]. We will examine these different mechanisms in the context of hematologic malignancies below, but it is also important to note that several studies have also highlighted the link between MSCs and metastatic process in solid tumors. In breast cancer, MSC activity through CCL5 release and Tac1 upregulation markedly increased tumoral metastatic capacity[13,14]. In neuroblastoma, differences in both qualitative and quantitative features of MSCs affect tumoral progression in BM[15]. A MSC subpopulation expressing stemness, endothelial and pericytic cell markers seems to impair neoplastic cells homing to BM in breast and prostate cancer models[16]. These findings, even if impossible to apply to hematologic malignancies, demonstrate that MSCs are implicated in the regulation of the interactions between neoplastic cells and BM niche.

References:
1. Tormin, A.; Brune, J.C.; Olsson, E.; Valcich, J.; Neuman, U.; Olofsson, T.; Jacobsen, S.E.; Scheding, S. Characterization of bone marrow-derived mesenchymal stromal cells (MSC) based on gene expression profiling of functionally defined MSC subsets. Cytotherapy 2009, 11, 114-128, doi:10.1080/14653240802716590.
2. Kuçi, S.; Kuçi, Z.; Kreyenberg, H.; Deak, E.; Pütsch, K.; Huenecke, S.; Amara, C.; Koller, S.; Rettinger, E.; Grez, M., et al. CD271 antigen defines a subset of multipotent stromal cells with immunosuppressive and lymphohematopoietic engraftment-promoting properties. Haematologica 2010, 95, 651-659, doi:10.3324/haematol.2009.015065.
3. Flores-Figueroa, E.; Varma, S.; Montgomery, K.; Greenberg, P.L.; Gratzinger, D. Distinctive contact between CD34+ hematopoietic progenitors and CXCL12+ CD271+ mesenchymal stromal cells in benign and myelodysplastic bone marrow. Laboratory investigation; a journal of technical methods and pathology 2012, 92, 1330-1341, doi:10.1038/labinvest.2012.93.
4. Li, H.; Ghazanfari, R.; Zacharaki, D.; Ditzel, N.; Isern, J.; Ekblom, M.; Méndez-Ferrer, S.; Kassem, M.; Scheding, S. Low/negative expression of PDGFR-α identifies the candidate primary mesenchymal stromal cells in adult human bone marrow. Stem cell reports 2014, 3, 965-974, doi:10.1016/j.stemcr.2014.09.018.
5. Tormin, A.; Li, O.; Brune, J.C.; Walsh, S.; Schütz, B.; Ehinger, M.; Ditzel, N.; Kassem, M.; Scheding, S. CD146 expression on primary nonhematopoietic bone marrow stem cells is correlated with in situ localization. Blood 2011, 117, 5067-5077, doi:10.1182/blood-2010-08-304287.
6. Maijenburg, M.W.; Kleijer, M.; Vermeul, K.; Mul, E.P.; van Alphen, F.P.; van der Schoot, C.E.; Voermans, C. The composition of the mesenchymal stromal cell compartment in human bone marrow changes during development and aging. Haematologica 2012, 97, 179-183, doi:10.3324/haematol.2011.047753.
7. Duscher, D.; Rennert, R.C.; Januszyk, M.; Anghel, E.; Maan, Z.N.; Whittam, A.J.; Perez, M.G.; Kosaraju, R.; Hu, M.S.; Walmsley, G.G., et al. Aging disrupts cell subpopulation dynamics and diminishes the function of mesenchymal stem cells. Scientific reports 2014, 4, 7144, doi:10.1038/srep07144.
8. Khong, S.M.L.; Lee, M.; Kosaric, N.; Khong, D.M.; Dong, Y.; Hopfner, U.; Aitzetmüller, M.M.; Duscher, D.; Schäfer, R.; Gurtner, G.C. Single-Cell Transcriptomics of Human Mesenchymal Stem Cells Reveal Age-Related Cellular Subpopulation Depletion and Impaired Regenerative Function. Stem cells (Dayton, Ohio) 2019, 37, 240-246, doi:10.1002/stem.2934.
9. Isern, J.; García-García, A.; Martín, A.M.; Arranz, L.; Martín-Pérez, D.; Torroja, C.; Sánchez-Cabo, F.; Méndez-Ferrer, S. The neural crest is a source of mesenchymal stem cells with specialized hematopoietic stem cell niche function. eLife 2014, 3, e03696, doi:10.7554/eLife.03696.
10. Wang, X.; Zou, X.; Zhao, J.; Wu, X.; E, L.; Feng, L.; Wang, D.; Zhang, G.; Xing, H.; Liu, H. Site-Specific Characteristics of Bone Marrow Mesenchymal Stromal Cells Modify the Effect of Aging on the Skeleton. Rejuvenation research 2016, 19, 351-361, doi:10.1089/rej.2015.1766.
11. Lee, M.W.; Ryu, S.; Kim, D.S.; Lee, J.W.; Sung, K.W.; Koo, H.H.; Yoo, K.H. Mesenchymal stem cells in suppression or progression of hematologic malignancy: current status and challenges. Leukemia 2019, 33, 597-611, doi:10.1038/s41375-018-0373-9.
12. Galland, S.; Stamenkovic, I. Mesenchymal stromal cells in cancer: a review of their immunomodulatory functions and dual effects on tumor progression. The Journal of pathology 2020, 250, 555-572, doi:10.1002/path.5357.
13. Karnoub, A.E.; Dash, A.B.; Vo, A.P.; Sullivan, A.; Brooks, M.W.; Bell, G.W.; Richardson, A.L.; Polyak, K.; Tubo, R.; Weinberg, R.A. Mesenchymal stem cells within tumour stroma promote breast cancer metastasis. Nature 2007, 449, 557-563, doi:10.1038/nature06188.
14. Corcoran, K.E.; Trzaska, K.A.; Fernandes, H.; Bryan, M.; Taborga, M.; Srinivas, V.; Packman, K.; Patel, P.S.; Rameshwar, P. Mesenchymal stem cells in early entry of breast cancer into bone marrow. PloS one 2008, 3, e2563, doi:10.1371/journal.pone.0002563.
15. Hochheuser, C.; van Zogchel, L.M.J.; Kleijer, M.; Kuijk, C.; Tol, S.; van der Schoot, C.E.; Voermans, C.; Tytgat, G.A.M.; Timmerman, I. The Metastatic Bone Marrow Niche in Neuroblastoma: Altered Phenotype and Function of Mesenchymal Stromal Cells. Cancers 2020, 12, doi:10.3390/cancers12113231.
16. Rossnagl, S.; Ghura, H.; Groth, C.; Altrock, E.; Jakob, F.; Schott, S.; Wimberger, P.; Link, T.; Kuhlmann, J.D.; Stenzl, A., et al. A Subpopulation of Stromal Cells Controls Cancer Cell Homing to the Bone Marrow. Cancer research 2018, 78, 129-142, doi:10.1158/0008-5472.Can-16-3507.

Round 2

Reviewer 1 Report

All my comments were addressed.